# Delineating the *Tnt1* Insertion Landscape of the Model Legume *Medicago truncatula* cv. R108 at the Hi-C Resolution Using a Chromosome-Length Genome Assembly

**DOI:** 10.3390/ijms22094326

**Published:** 2021-04-21

**Authors:** Parwinder Kaur, Christopher Lui, Olga Dudchenko, Raja Sekhar Nandety, Bhavna Hurgobin, Melanie Pham, Erez Lieberman Aiden, Jiangqi Wen, Kirankumar S Mysore

**Affiliations:** 1UWA School of Agriculture and Environment, The University of Western Australia, Crawley, WA 6009, Australia; erez.Lieberman@bcm.edu; 2The Center for Genome Architecture, Department of Molecular and Human Genetics, Baylor College of Medicine, Houston, TX 77030, USA; christopher.g.lui@gmail.com (C.L.); olga.Dudchenko@bcm.edu (O.D.); melanie_pham@ymail.com (M.P.); 3Center for Theoretical and Biological Physics, Departments of Computer Science and Computational and Applied Mathematics, Rice University, Houston, TX 77030, USA; 4Noble Research Institute, LLC., Ardmore, OK 73401, USA; rsnandety@noble.org (R.S.N.); jwen@noble.org (J.W.); 5La Trobe Institute for Agriculture and Food, Department of Animal, Plant and Soil Sciences, School of Life Sciences, AgriBio Building, La Trobe University, Bundoora, VIC 3086, Australia; bhavna.hurgobin@hotmail.com; 6Australian Research Council Research Hub for Medicinal Agriculture, AgriBio Building, La Trobe University, Bundoora, VIC 3086, Australia; 7Broad Institute of MIT and Harvard, Cambridge, MA 02139, USA; 8Shanghai Institute for Advanced Immunochemical Studies, ShanghaiTech, Pudong 201210, China

**Keywords:** Leguminosae, *Medicago truncatula* cv. R108, HiC, chromosome-length genome assembly, *Tnt1* insertion landscape

## Abstract

Legumes are of great interest for sustainable agricultural production as they fix atmospheric nitrogen to improve the soil. *Medicago truncatula* is a well-established model legume, and extensive studies in fundamental molecular, physiological, and developmental biology have been undertaken to translate into trait improvements in economically important legume crops worldwide. However, *M. truncatula* reference genome was generated in the accession Jemalong A17, which is highly recalcitrant to transformation. *M. truncatula* R108 is more attractive for genetic studies due to its high transformation efficiency and *Tnt1*-insertion population resource for functional genomics. The need to perform accurate synteny analysis and comprehensive genome-scale comparisons necessitates a chromosome-length genome assembly for *M. truncatula* cv. R108. Here, we performed in situ Hi-C (48×) to anchor, order, orient scaffolds, and correct misjoins of contigs in a previously published genome assembly (R108 v1.0), resulting in an improved genome assembly containing eight chromosome-length scaffolds that span 97.62% of the sequenced bases in the input assembly. The long-range physical information data generated using Hi-C allowed us to obtain a chromosome-length ordering of the genome assembly, better validate previous draft misjoins, and provide further insights accurately predicting synteny between A17 and R108 regions corresponding to the known chromosome 4/8 translocation. Furthermore, mapping the *Tnt1* insertion landscape on this reference assembly presents an important resource for *M. truncatula* functional genomics by supporting efficient mutant gene identification in *Tnt1* insertion lines. Our data provide a much-needed foundational resource that supports functional and molecular research into the Leguminosae for sustainable agriculture and feeding the future.

## 1. Introduction

Sustainable agricultural production involves growing food with low fertilizer input without damaging the underlying soil [1]. Legumes are of great interest for sustainable agriculture because they produce nitrogen via symbiotic nitrogen fixation, improving soil health [2,3]. Most legumes, however, have large/complex genomes and are outcrossing species, making genetic studies difficult. *Medicago truncatula* was chosen as a model legume due to its small genome [4], diploidy, autogamy, and short life cycle. In the past two decades, extensive studies have been undertaken in plant–bacterial symbioses and fundamental molecular, physiological, and developmental biology of *M. truncatula* to translate and improve traits in economically important legume crops [5,6,7,8,9]. The release of the *M. truncatula* accession Jemalong A17 reference genome sequence and generation of the *Tnt1*-based insertion mutant population for accession R108 have greatly accelerated functional genomics studies in *M. truncatula* [10,11,12]. The *M. truncatula* reference genome was generated in A17, which is highly recalcitrant to transformation, whereas the *Tnt1* mutant population was generated in R108, with a much higher transformation efficiency. Phylogenetically, R108 is one of the most distant *M. truncatula* accessions from A17 [13]. R108 is more attractive for genetic studies due to its high transformation efficiency [10]. Recently, R108 has become popular in legume research communities with its near-saturated *Tnt1*-insertion population, which is widely used in most areas of legume functional genomic analysis [10,12]. The *Tnt1* insertion population comprises 21,700 regenerated lines, encompassing more than a half-million randomly distributed *Tnt1* insertions [12]. Due to the lack of high-quality pseudomolecules (chromosomes) in R108, all *Tnt1* insertions are mapped to the A17 genome. However, compared to R108 and other *M. truncatula* genotypes, A17 has a large (~30 Mb) reciprocal translocation between chromosomes 4 and 8 [4], resulting in inaccurate synteny analysis between *M. truncatula* and other legume genomes and aberrant recombination in genetic crosses, including crosses between A17 and R108 [14]. In addition, evolutionary whole-genome duplications [13,15] and frequent local duplications make genome assembly difficult. Therefore, having two high-quality references in *M. truncatula* will allow us to perform more accurate synteny analysis and comprehensive genome-scale comparisons, and calls for the *M. truncatula* cv. R108 genome sequence.

Three years ago, the first draft assembly of *M. truncatula* cv. R108 was constructed using a combination of PacBio, Dovetail, and BioNano technologies, as described by Moll et al. (2017). Recently, we and others significantly improved draft genomes using data derived from in situ Hi-C [16,17,18,19]. As Hi-C can estimate the relative proximity of loci in the nucleus, Hi-C contact maps can be used to correct misjoins, anchor, order, and orient contigs and scaffolds. This process improves contig accuracy and typically yields chromosome-length scaffolds. To broaden the range of genetic resources available for the model legume *M. truncatula*, we used Hi-C to improve the R108 v1.0 draft assembly, producing a genome assembly for *M. truncatula* cv. R108 with chromosome-length scaffolds. Approximately 387,000 flanking sequence tags (FSTs), identified from approximately 21,000 *Tnt1* insertion lines of *M. truncatula* cv. R108, were mapped onto pseudo-chromosomes of R108.

## 2. Results

### 2.1. Assembly of M. truncatula Accession R108 with Chromosome-Length Scaffolds

The first draft assembly of *M. truncatula* cv. R108 was constructed using a combination of PacBio, Dovetail, and BioNano technologies [20]. The resulting assembly (R108 v1.0) comprised 402 Mb of sequence (contig N50 length: 5.93 Mb) partitioned among 909 scaffolds. Here, we generated in situ Hi-C data [16,18] from *M. truncatula* cv. R108 leaves to improve its initial draft assembly [19,21]. Scaffolds/contigs shorter than 1 Kb were not anchored from the R108 v1.0 assembly; the remaining scaffolds were anchored, ordered, oriented, and corrected for misjoins using the Hi-C data. After manual refinement using Juicebox Assembly Tools [21], as shown in Figure 1, the resulting assembly, named MedtrR108_hic, was represented by 801 scaffolds, of which eight were chromosome-length scaffolds (N50 length of 51.86 Mb), ranging from 37.80 to 55.90 Mb. The chromosome-length scaffolds spanned 97.62% of the sequenced bases in the entire assembly. The remaining 793 scaffolds (N50 length of 18.96 Kb) constituted the remaining 2.38% of the total assembly. The circular snail plots describing the assembly statistics of MedtrR108_hic and R108 v1.0 are shown in Figure 2a,b. These results are further summarized in Table 1. Additional assembly statistics can be found in Appendix A.

Assessment of the Hi-C genome assembly quality was performed via CEGMA analysis [22] to identify the presence of core eukaryotic genes (CEGs). From the 248 CEGs analyzed, 228 (91.94%) complete genes and 245 (98.79%) partial genes were identified in the Hi-C assembly (Appendix A). In contrast, only 223 (89.92%) and 242 (97.52%) complete and partial CEGs were found in the R108 v1.0 assembly (Appendix A).

### 2.2. Genome Annotation and Functional Characterization

Reannotation of the MedtrR108_hic genome assembly predicted 39,027 high-confidence, protein-coding genes, which is lower than the 55,706 and 44,623 protein-coding genes annotated in the R108 v1.0 (GenBank accession no. GCA_002024945.1) and A17 Mt5.0 assemblies (GenBank accession no. GCA_003473485.2), respectively [14,20]. However, assessment of gene space completeness via a Benchmarking Universal Single-Copy Orthologs (BUSCO) analysis [23] showed that the MedtrR108_hic assembly harbored a higher percentage of complete BUSCOs (96.73%) than the R108 v1 assembly (91.94%) among the 2326 BUSCO groups searched (Appendix A). The percentages of fragmented and missing BUSCOs were also less in the MedtrR108_hic assembly than the R108 v1.0 assembly. Further, the number of complete BUSCOs (single copy and duplicated) were more comparable between MedtrR108_hic and A17 Mt5.0 than between R108 v1 and A17 Mt5.0.

Of the 39,027 genes, 36,994 (94.79%) had at least one hit against either A17 Mt5.0 proteins [14], TAIR10 (https://www.arabidopsis.org/, accessed on 17 March 2021), Phytozome [24] v13, Swissprot [25], RefSeq [26] or TrEMBL [25] databases, as revealed by the BLAST searches (Appendix A). Phytozome v13 and TrEMBL were the most informative databases, assigning functional annotations to 93.53% and 94.20% of the predicted genes, respectively. The R108 v1.0 genes had lower percentages of BLAST hits overall (84.72%), further confirming the relatively more complete annotation of the MedtrR108_hic assembly compared to the R108 v1.0 assembly (Appendix A). 

The MedtrR108_hic protein-coding genes were also mined for protein domains and annotated with gene ontology (GO) terms. A total of 29,504 (75.60%) and 19,086 (48.90%) genes had at least one protein domain and GO term assigned to them, respectively (Appendix A). All publicly available RNA-Seq accessions used for annotation are presented as Appendix A.

### 2.3. Synteny Analysis and Chromosomal Translocation

A high degree of collinearity was observed between A17 Mt5.0 and MedtrR108_hic (Figure 3a,b). However, as reported previously [14,27], translocation between chromosomes 4 and 8 of the A17 genome was visible. More precisely, a 12 Mb syntenic region was observed between A17 Mt5.0 chromosome 4 (46.93–64.75 Mb) and MedtrR108_hic chromosome 8 (32.86–50.23 Mb). Further, a 17 Mb syntenic region between A17 Mt5.0 chromosome 8 (37.03–49.69 Mb) and MedtrR108_hic chromosome 4 (41.12–35.19 Mb) was visible. While the 12 Mb syntenic region was also reported in A17 Mt5.0 versus R108 v1.0 [14], the 17 Mb syntenic region observed in the current study is significantly larger than the 4 Mb syntenic region reported between R108 v1.0 chromosome 8 and A17 Mt5.0 chromosome 4 [14]. Furthermore, Pecrix et al. identified three additional translocations in A17 Mt5.0R108 v1.0, but these were absent from the A17 Mt5.0 versus MedtrR108_hic comparison. As previously reported [14], inversion in the first 8.7 Mb region of chromosome 1 of A17 Mt5.0 was also clearly visible [14].

Overall, the A17 Mt5.0 versus MedtrR108_hic syntenic genes could be arranged in a smaller number of larger blocks than the A17 Mt5.0 versus R108 v1.0 syntenic genes. A total of 25,548 syntenic genes were identified between A17 Mt5.0 and MedtrR108_hic (Appendix A), which could be arranged in 59 collinear blocks. The largest block (no. 54) contained 2574 genes, while the smallest block (no. 49) contained four genes (Appendix A). In contrast, 26,348 syntenic genes were identified between A17 Mt5.0 and R108 v1.0 (Appendix A), which could be arranged into 121 collinear blocks. The largest block (no. 66) contained 1535 genes, while the smallest block (no. 73) contained four genes (Appendix A). 

Of the 25,548 syntenic genes identified in A17 Mt5.0 versus MedtrR108_hic, 2676 (10.47%) genes were found in the translocated regions (Appendix A); of which, 1143 were found in the 12 Mb region, with the remaining 1533 genes in the 17 Mb region (Appendix A). Of the 26,348 syntenic genes identified between A17 Mt5.0 and R108 v1.0, 1590 (6.03 %) genes were found in the translocated regions; most of which (1159) were found in the 12 Mb region, while the remaining 431 genes were in the 4 Mb region (Appendix A). The GO terms commonly enriched in the translocated regions between A17 Mt5.0 and MedtrR108_hic (Appendix A) and A17 Mt5.0 and R108 v1.0 (Appendix A) comprised the following stress-response related-terms: response to water deprivation, plant-type hypersensitive response, response to ethylene, response to abscisic acid, and response to jasmonic acid (Appendix A).

### 2.4. Mapping Tnt1 Insertion Sites in the M. truncatula R108 Hi-C Genome Assembly

From the 21,741 *Tnt1* insertion lines generated in *M. truncatula* cv. R108, 392,396 FSTs were recovered using TAIL-PCR and Sanger or Illumina sequencing [12]. The average sequence length of these FSTs is 363 bp. To identify the signature sequence in FSTs, we processed all FSTs and obtained 221,275 high-confidence FST sequences. The remaining 171,121 FSTs that lacked signature sequence likely resulted from AD primer end sequencing. Of the 221,275 FSTs, 202,788 (92%) were successfully mapped to the *M. truncatula* R108 reference genome MedtrR108_hic with an identity greater than 90% (Table 2). A total of 201,427 *Tnt1* insertions were mapped to eight chromosomes, with an average of 25,178 insertions per chromosome (Table 2). The most *Tnt1* insertions (27,902; 12.6%) were mapped onto chromosome 1 and the least (16,433; 7.4%) were mapped onto chromosome 6 (Table 2). It is reasonable to observe low numbers on chromosome 6 as it is the smallest of the eight chromosomes. In addition, 1361 *Tnt1* insertions were mapped onto the unanchored scaffolds. The mapping of *Tnt1* insertions across all R108 chromosomes confirms previous results that showed random *Tnt1* insertions based on the *M. truncatula* A17 genome [12]. All *Tnt1* insertions were mapped to chromosomes in the Hi-C assembly of the R108 genome based on physical chromosome location through circos genome plots (Figure 4).

### 2.5. Comparison of Tnt1 Insertions Using M. truncatula R108 Hi-C or A17 v5.0 Genic Regions and Functional Annotation of Genes with Insertions

In the *M. truncatula* A17 v5.0 reference genome [14], 44,624 genes were predicted and annotated. Our Hi-C assembly predicted 39,027 genes in *M. truncatula* R108. From 202,788 high-confidence FSTs (Appendix A), there were 24,052 genes with exact *Tnt1* insertion sites (61.62%) in the R108 Hi-C assembly (Appendix A). We found a similar percentage of genes (60%; 26,717 genes) in the *M. truncatula* A17 v5.0 reference genome with *Tnt1* insertion in at least one gene (Appendix A). A list of the GO annotations analyzed for genes with *Tnt1* insertions and the gene groups are summarized in Appendix A. In the R108 Hi-C version, there were at least 19,008 genes (48.7%) with more than one *Tnt1* insertion, contrasting with 18,352 genes (41.12%) in the *M. truncatula* A17 v5.0 reference genome (Appendix A). There were at least 12,746 genes (32.65%) with at least four *Tnt1* insertions in the R108 Hi-C assembly, contrasting with 22.29% of the genes (9949 genes) in the *M. truncatula* A17 v5.0 reference genome (Appendix A). An average of 4.07 *Tnt1* insertions per gene was observed in the MedtrR108_hic assembly compared to 4.33 insertions per gene in *M. truncatula* A17 v5.0 (Appendix A).

The most frequently hit gene when *M. truncatula* A17 v5.0 genome was used for analysis is MtrunA17Chr5g0441701 (putative peptidyl prolyl isomerase), with 135 *Tnt1* insertions (Appendix A), while the two genes with more *Tnt1* insertions when MedtrR108_hic assembly was used for analysis are MedtrR108_hic. Hi-C_scaffold_8.3452 (Eukaryotic and viral aspartyl proteases active site protein) and MedtrR108_hic. Hi-C_scaffold_2.2064 (RHN73856.1 putative FAS1 domain-containing protein) with 143 and 139 *Tnt1* insertions, respectively (Appendix A). The genes that did not have insertions were also identified (Appendix A). It is reasonable to assume that *Tnt1* in the existing insertion population disrupts majority of genes in the *M. truncatula* genome. GO ontology and annotation were performed for all genes with frequent insertions and insertions into genes with less frequency (Appendix A).

AgriGO v2.0 [28] analysis was used to enrich the frequently inserted 7737 genes in GO categories, which were selected based on genes that are inserted more than the average insertion number (i.e., 4.33 insertions per gene). The results showed that these frequently inserted genes fall into the following five pathways: stress, signaling, secondary metabolism, transport, and nucleotide metabolism (Appendix A). The significant GO terms under the biological processes are response to stress, response to stimulus, defense response, protein phosphorylation, and transmembrane transport (Appendix A). The significant GO terms under molecular functions are ATP binding, active transmembrane transporter activity, protein tyrosine kinase activity, and transporter activity (Appendix A). The GO enrichment analysis revealed similar results to the pathway analysis and corresponded with the previously reported data [12].

### 2.6. Tnt1 Insertions in Genes in the Syntenic Regions

Syntenic regions between A17 v5.0 and R108 v1.0 genome were obtained from the publicly released v1.0 [14]. The syntenic region between A17 v5.0 and MedtrR108_hic syntenic genes could be arranged into a smaller number of larger blocks than the A17 Mt5.0 versus R108 v1.0 syntenic genes. A total of 25,548 syntenic genes were identified between A17 Mt5.0 and MedtrR108_hic (Appendix A), which could be arranged in 59 collinear blocks. The largest block (no. 54) contained 2574 genes, while the smallest block (no. 49) contained four genes. We identified 17,766 genes present in all syntenic blocks combined between A17 and R108 (Appendix A). Each of the *Tnt1* genic insertions in the syntenic regions and the GO annotation is presented in Appendix A. Individual gene numbers from each block are identified and presented as a Appendix A. Six syntenic blocks (54, 25, 9, 36, 12, and 31) have more than 1000 genes with *Tnt1* insertions (Appendix A). The highest number of genic *Tnt1* insertions are in Block 54 with 1787 genes (Appendix A).

## 3. Discussion

The MedtrR108_hic assembly is a significant improvement on the R108 v1.0 assembly, with its smaller number of larger scaffolds, higher scaffold N50 value and improved CEGMA results. While fewer genes were annotated in the Hi-C assembly, the gene content appeared to be more complete than the R108 v1.0 annotation, as reflected in the BLAST and BUSCO results for MedtrR108_hic [20] processed through the MAKER-P pipeline [29] for annotation; only ab initio gene predictions from RNA-Seq alignments were used as the source of evidence. In the current study, a combination of ab initio gene predictions from RNA-Seq alignments and evidence from protein homology studies were used for annotation via the BRAKER2 [30] pipeline and EvidenceModeler [31]. The latter tool primarily leverages the ab initio predictions as its source of gene model components, and then leverages the protein and transcript alignment data to guide its choice of best models. Therefore, any ab initio predictions not supported by the protein/transcript alignments are discarded. This strict filtering could explain why we observed a reduction in the number of genes. Additionally, the RNA-Seq libraries used to annotate the Hi-C assembly were derived from root tissue [20] and leaf tissue (data generated in-house). However, the R108 v1.0 assembly was annotated using RNA-Seq data from root tissue only [20]. This could explain why the current annotation is more complete. 

The abnormal conformation of chromosomes 4 and 8 in genotype A17 is well-known [14,27]. The smaller number of larger collinear blocks identified between A17 Mt5.0 and MedtrR108_hic, coupled with the larger 17 Mb translocation between A17 Mt5.0 chromosome 4 and MedtrR108_hic chromosome 8, reflects the more contiguous nature of the Hi-C assembly than R108 v1.0. Furthermore, the absence of the three additional breakpoints (BKPT 2, 3, and 4) identified by Pecrix et al. in the Hi-C assembly when comparing A17 Mt5.0 and R108 v1.0 suggests that these breakpoints occurred as a result of the more fragmented nature of the R108 v1.0 assembly or the presence of errors in the assembly. Therefore, it is unlikely that these breakpoints represent true structural variations in A17. On the other hand, the inversion in A17 Mt5.0 for both MedtrR108_hic and R108 v1.0 indicates that this structural variation is real and constitutes a second distinctive structural feature of the A17 genotype. This inversion was also visible when A17 Mt5.0 was compared with the genetic maps of *Medicago sativa* and *Pisum sativa*, the species most closely related to *M. truncatula* [14].

*Tnt1* insertion lines have become more and more popular due to their powerful, versatile applications in forward and reverse genetics. The *Tnt1* lines were generated in the R108 background due to its high transformation and regeneration efficiency. The A17 and R108 genomes significantly differed due to their phylogenetic distance [32]. Though most genes in both genomes have high similarity, there are a significant number of genes that have moderate similarity, which will cause ambiguity in determining whether a BLAST search result of a gene with the A17 sequence is a true hit in the *Tnt1* FST database. Therefore, a high-quality R108 genome assembly was needed. Compared to the genome R108 v1.0, the assembly quality of MedtrR108_hic has significantly improved, especially in the syntenic translocation regions, where *Tnt1* FST mapping is more accurate in the MedtrR108_hic genome.

Genome-editing technology, especially Clustered Regularly Interspaced Short Palindromic Repeat/CRISPR-associated protein 9 (CRISPR/Cas9) technology, has become more powerful and applicable to many plant species, including *M. truncatula*. CRISPR/Cas9 is an innovative technology, offering excellent opportunities for plant genetics and functional genomics research. Its advantages include target specificity, effectiveness, precision, and feasibility for multiple genome manipulation options [33]. Accurate plant gene sequences are critical for gene editing. The improved genome editing efficiency in *M. truncatula* [34] should increase CRISPR/Cas9 technology use. Due to significant differences in the transformation efficiencies between A17 and R108, R108 is the first choice for genome editing practices. The improved genome assembly of R108 provides a solid foundation for future genome editing research in the legume community.

## 4. Materials and Methods

### 4.1. Hi-C Library Preparation and Sequencing

In situ Hi-C was performed as described previously [18] using frozen leaves from *Medicago truncatula* cv. R108. Briefly, frozen leaf tissue was crosslinked, ground and then lysed with nuclei permeabilized but still intact. DNA was then restricted with *Mbo*I restriction enzyme and the overhangs filled in incorporating a biotinylated base. Free ends were then ligated together in situ. Crosslinks were reversed, the DNA was sheared to 300–500 bp and then biotinylated ligation junctions were recovered with streptavidin beads.

Standard Illumina library construction protocol was used for DNA sequencing. Briefly, DNA was end-repaired using a combination of T4 DNA polymerase, *Escherichia coli* DNA Pol I large fragment (Klenow polymerase), and T4 polynucleotide kinase. The blunt, phosphorylated ends were treated with Klenow fragment (3′ to 5′ exo minus) and dATP to yield a protruding 3- ‘A’ base for ligation of Illumina’s adapters which have a single ‘T’ base overhang at the 3’ end. After adapter ligation, DNA was PCR amplified with Illumina primers for 14 cycles and library fragments of 400–600 bp (insert plus adaptor and PCR primer sequences) were purified using SPRI beads. The purified DNA was captured on an Illumina flow cell for cluster generation. Libraries were sequenced on the NextSeq500 following the manufacturer’s protocols. The same R108 lineage used for generating *Tnt1* insertion lines was used for Hi-C. The resulting library was sequenced to yield approximately 48× coverage of the *M. truncatula* genome.

### 4.2. Genome Assembly

The Hi-C library was processed against the R108 v1.0 genome assembly [20] the Juicer pipeline [35]. The assembly was performed as described [19,21]. Briefly, after excluding scaffolds shorter than 1 Kb, the 3D De Novo Assembly (3D-DNA) pipeline was run using the in situ Hi-C data to anchor, order, orient, and correct misjoins in the R108 v1.0 scaffolds. Lastly, a manual refinement step was performed using Juicebox Assembly Tools [21]. The resulting contact maps were visualized using the 3D-DNA and Juicebox visualization system [19,21,36].

### 4.3. Genome Annotations

A multistep approach consisting of ab initio gene predictions, protein alignments, and transcript assembly was used to annotate the MedtrR108_hic reference genome assembly. RepeatModeler [37] v1.0.9. was used to identify interspersed repeats and low complexity DNA sequences in the assembly. These regions were soft-masked in the assembly using RepeatMasker [38] v4.0.8. The resulting soft-masked assembly was used as the input for BRAKER2 [30] for ab initio gene prediction using GeneMark-ET v4.33 [39] and AUGUSTUS v3.3.1 [40] based on alignments of RNA-Seq data. Prior to alignment, adapter sequences and low-quality bases were trimmed from the Illumina RNA-Seq libraries using Trimmomatic [41] v0.33. (sliding window, minimum quality score: 20). Trimmed libraries were aligned to MedtrR108_hic reference assembly using HISAT2 [42] v2.1.0. (insert size 0 to 1000). The resulting SAM files were converted to BAM format using SAMtools [43] v1.3, which were then merged prior to transcript assembly using Stringtie [44] v 1.3.5. 

A total of 89,910 *M. truncatula* protein sequences were accessed and downloaded from NCBI (https://www.ncbi.nlm.nih.gov/, accessed on 4 April 2019). These sequences were aligned to the soft-masked reference using Exonerate [45] v2.2.0 (--softmasktarget --model protein2genome --showvulgar no --showalignment no --showquerygff no --showtargetgff yes --percent 80).

Finally, EvidenceModeler [31] v1.1.1. was used to combine the gene predictions from GeneMark-ET and AUGUSTUS, protein alignments from Exonerate, and the assembled transcripts from Stringtie to obtain the final gene set. 

### 4.4. Assessment of Genome Assembly and Annotation Quality

Assessment of the Hi-C genome assembly quality and completeness was performed via CEGMA [22] v2.5. to identify the presence of CEGs. BUSCO [23] v4.14. was run using the eudicotyledons_odb10 dataset in protein mode to evaluate the annotation quality.

### 4.5. Functional Annotation of the Predicted Genes

A protein BLAST search (blastp) was performed by aligning the predicted proteins to several databases using BLAST [46] v2.2.29 (minimum e-value 1e-5). The databases used were the A17 Mt5.0 proteins [14], SwissProt [25], TrEMBL [37], TAIR10 (https://www.arabidopsis.org/, accessed on 17 March 2021), RefSeq [26], and Phytozome [24] v13, accessed and downloaded on 30 November 2020. For each predicted protein, the hit with the highest score and lowest e-value was chosen as annotation. KEGG numbers were assigned to all predicted proteins using BLASTKOALA (taxonomy group: Plants, KEGG GENES database: family_eukaroytes) [47]. The predicted genes were annotated with GO terms and mined for protein domains using InterProScan [48] v5.45–80.0 and the following databases: TIGRFAM, ProDom, PANTHER, Pfam, PrositeProfiles, PrositePatterns, Coils, SUPERFAMILY, SFLD, SMART, PRINTS, MobiDBLite, and PIRSF.

### 4.6. Genome Alignment and Detection of Chromosomal Translocation

Whole genome alignments were performed between MedtrR108_hic and A17 Mt5.0 (GenBank Accession no. GCA_003473485.2) using minimap 2 [32] v2.17 (-x asm5). The alignments were filtered [14] (primary alignments only [tp:A:P]; alignment block length > 10 Kb; approximate per-base sequence divergence (dv) score lower than 0.8), and visualized using D-genies [49] v1.2.0 (dot plot) and Circos [50] v0.69 (circular genome plot). 

The python version of MCScan [51] was used to identify syntenic regions and their corresponding genes between MedtrR108_hic and A17 Mt.5.0, and R108 v1.0 (GenBank Accession no. GCA_002024945.1) and A17 Mt5.0. A syntenic region was defined as one with at least 10 shared genes (--(-minspan=10).

### 4.7. Mapping of Tnt1 Insertion Lines and Functional Gene Group Analysis

To accurately identify *Tnt1* insertion sites in the *M. truncatula* genome, all FST sequences shorter than 50 bp, without the *Tnt1* signature sequence (‘CCCAACA,’ ‘CATCATCA’ or ‘TGATGATGTCC’), or the *Tnt1* signature sequence outside the 28 bp from the beginning or end of the FST sequence were discarded. The preprocessed reliable FST sequences were aligned to the *M. truncatula* A17 version4 (Mt4.0) or version5 (Mt5.0) and R108 Hi-C assembly reference genomes using BLASTN with an e-value threshold ≤1.00 × 10^−5^. The FST sequences with best hit from BLAST analysis were further processed for downstream analysis. Only hits with at least 90% sequence identity were considered and used for functional gene group analysis. Functional gene group analysis was performed as described elsewhere [12].

## 5. Conclusions

Using in situ Hi-C data, we improved the *M. truncatula* cv. R108 genome assembly by correcting misjoins and ordering and orienting scaffolds to generate eight chromosome-length large scaffolds that correspond to the eight chromosomes in the A17 reference genome. Compared to the previous version (v1.0) of the R108 genome, the newly assembled MedtrR108_hic genome is a significant improvement due to its smaller number of larger scaffolds, higher scaffold N50 value, and improved CEGMA results. MedtrR108_hic also provides insight into how to accurately predict syntenies in the chromosome 4/8 translocation regions between A17 and R108. Furthermore, mapping the *Tnt1* insertion landscape onto the current reference assembly provides a much-needed foundational resource for functional genomics studies in the legume community.

## 6. Patents

O.D., M.P., C.L, and E.L.A. are inventors on U.S. provisional patent application 62/347,605 filed 8 June 2016, by the Baylor College of Medicine and the Broad Institute, relating to the assembly methods in this manuscript.

## Figures and Tables

**Figure 1 ijms-22-04326-f001:**
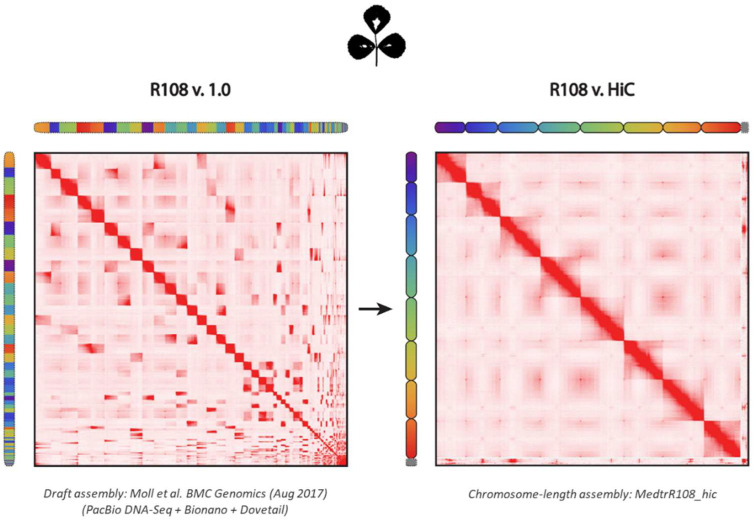
Hi-C map of the draft and chromosome-length assemblies of *Medicago truncatula* cv. R108 genome. Contact matrices were generated by aligning the same Hi-C data set to the R108 v1.0 draft genome (**left**) and MedtrR108_hic genome assembly generated using Hi-C (**right**). Pixel intensity in the matrix indicates how often a pair of loci co-locate in the nucleus. Correspondence between loci in the draft and final assemblies is illustrated using chromograms. The chromosome-length assembly scaffolds in Med-trR108_hic are assigned a linear color gradient. hic are assigned a linear color gradient; the same colors are then used for the corresponding loci in the R108v1.0 (**left**). The draft scaffolds are ordered by sequence name. Gridlines highlight the boundaries of eight chromosome-length scaffolds in MedtrR108_hic (**right**). Scaffolds smaller than 10 kb in R108v1.0 are not included.

**Figure 2 ijms-22-04326-f002:**
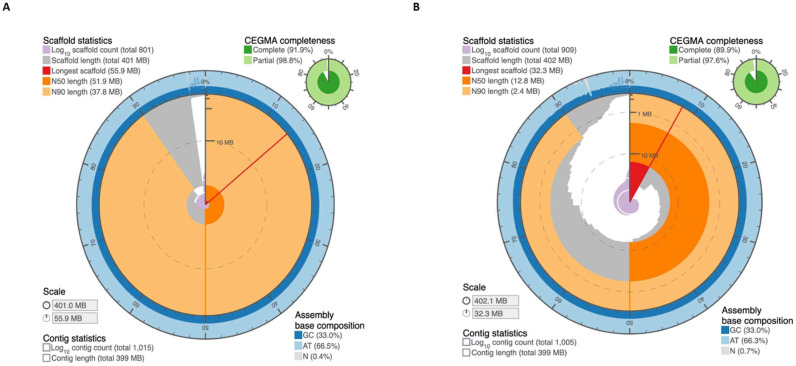
Snail plots describing the assembly statistics of the (**A**) MedtrR108_hic assembly and (**B**) R108 v1.0 assembly. Note the larger values for the longest scaffolds, N50 and N90, for MedtrR108_hic than R108 v1.0. The plots were generated using https://github.com/rjchallis/assembly-stats, accessed on 17 March 2021.

**Figure 3 ijms-22-04326-f003:**
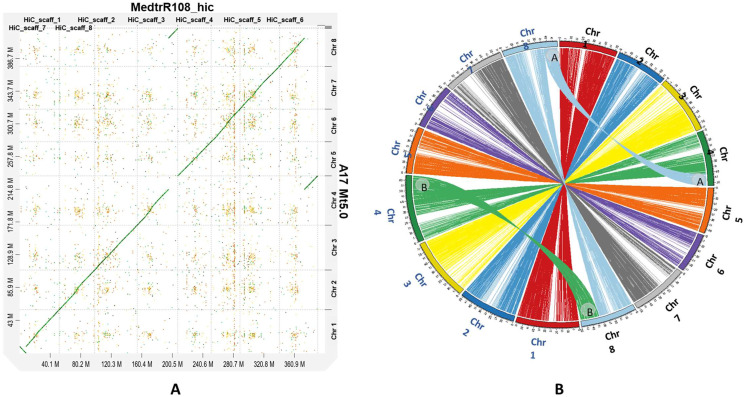
Assembly using Hi-C improves comparative analysis. (**A**) Whole-genome alignments of MedtrR108_hic versus A17 Mt5.0 highlight the peculiarity of the A17 genotype better than those between R108 v1.0 and A17 Mt5.0 [14]. (**B**) Circos plot depicts the genome structure of the syntenic relationship between MedtrR108_hic (chromosome names on right in black) and A17 Mt5.0 (chromosome names on left in blue) via syntenic links. The translocated regions on chromosomes 4 and 8 are highlighted: A denotes a 12 Mb syntenic region between MedtrR108_hic chromosome 4 (41.1–53.2 Mb) and A17 Mt5.0 chromosome 8 (37–49.7 Mb), and B denotes a 17 Mb syntenic region between MedtrR108_hic chromosome 8 (32.9–50.2 Mb) and A17 Mt5.0 chromosome 4 (46.9–64.7 Mb). The syntenic links represent syntenic blocks that are at least 50 Kbp long, and chromosome sizes are shown in Mb. Only the largest scaffolds/chromosomes determined syntenic relationships.

**Figure 4 ijms-22-04326-f004:**
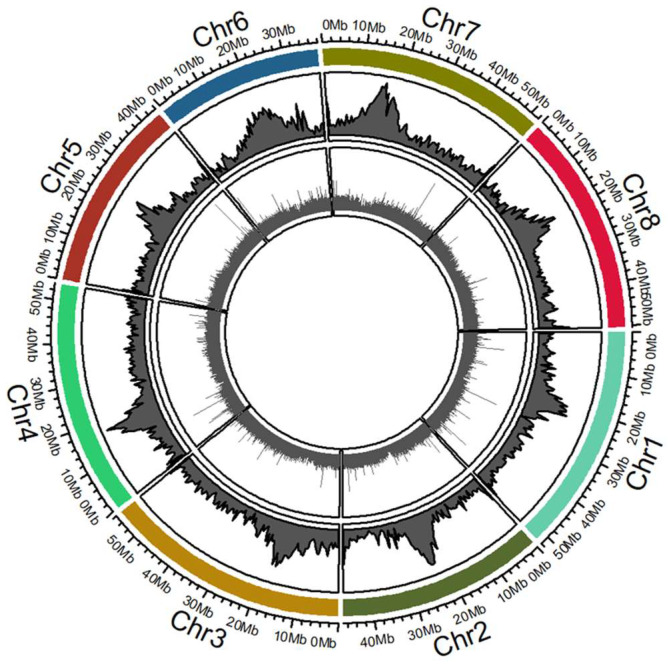
Circular genomic visualization of *Tnt1* insertions in *Medicago truncatula* R108 genome. The figure was generated using the R statistical platform in the Rcircos package. The outer band (outer circle) has chromosome locations (Chr1-Chr8). Each of the chromosome regions was divided into 500 Kb bins and plotted as bins with specific genomic locations. The first band of the circle represents the GC percentage of the chromosome regions specific to those divided bins. The second inner circle represents *Tnt1* insertions (as a measure of their FST lengths) in different chromosomes of the MedtrR108_hic assembly.

**Table 1 ijms-22-04326-t001:** Assembly statistics for the MedtrR108_hic genome assembly. Note that scaffolds smaller than 1 Kbp are excluded from the analysis.

Statistics	MedtrR108_hic
Draft scaffolds	
Base pairs	399,348,955
Number of contigs	1005
Contig N50	5,925,378
Number of scaffolds	909
Scaffold N50	12,848,239
Chromosome-length scaffolds	
Base pairs	390,045,474
Number of contigs	209
Contig N50	6,045,855
Number of scaffolds	8
Scaffold N50	51,860,634
Small scaffolds	
Base pairs	5,840,890
Number of contigs	248
Contig N50	24,000
Number of scaffolds	236
Scaffold N50	24,736
Tiny scaffolds	
Base pairs	3,462,591
Number of contigs	558
Contig N50	9246
Number of scaffolds	557
Scaffold N50	9246

**Table 2 ijms-22-04326-t002:** *Tnt1* insertion distribution on the *Medicago truncatula* R108 Hi-C genome.

Mapping Description	No of FSTs	% of Total FSTs
FSTs mapped to Chromosome 1	27,902	12.61
FSTs mapped to Chromosome 2	24,559	11.1
FSTs mapped to Chromosome 3	27,679	12.51
FSTs mapped to Chromosome 4	26,975	12.19
FSTs mapped to Chromosome 5	25,313	11.44
FSTs mapped to Chromosome 6	16,433	7.43
FSTs mapped to Chromosome 7	25,451	11.5
FSTs mapped to Chromosome 8	27,115	12.25
Total mapped to 8 chromosomes	201,427	91.03
Total mapped to non Chr scaffolds	1361	0.62

## Data Availability

All the data is available under the DNA Zoo BioProject accession PRJNA512907.

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
