# Peer review of "Delineating the Tnt1 Insertion Landscape of the Model Legume Medicago truncatula cv. R108 at the Hi-C Resolution Using a Chromosome-Length Genome Assembly"

_ijms, 2021, doi:10.3390/ijms22094326_

Round 1

Reviewer 1 Report

In this study, the authors performed in situ Hi-C to anchor, order, orient scaffolds, and correct misjoins of contigs in a previously published genome assembly of Medicago truncatula cv. R108. The Tnt1 insertion landscape was then characterized. Overall, this is a well-written and well-organized research article, with some interesting findings.

Although you cited a reference for the procedure of in situ Hi-C in Hi-C library preparation and sequencing, it would be better if you could briefly describe it in this manuscript instead of only referring to another paper.

It would be great if a flow chart should be provided for an easy understanding of how this entire study was performed.

Author Response

Response to Reviewer 1 Comments

In this study, the authors performed in situ Hi-C to anchor, order, orient scaffolds, and correct misjoins of contigs in a previously published genome assembly of Medicago truncatula cv. R108. The Tnt1 insertion landscape was then characterized. Overall, this is a well-written and well-organized research article, with some interesting findings.

Point 1: Although you cited a reference for the procedure of in situ Hi-C in Hi-C library preparation and sequencing, it would be better if you could briefly describe it in this manuscript instead of only referring to another paper.

Response 1: We have added the following text to the Materials & Methods section 4.1 as suggested.

Frozen plant tissue was crosslinked, ground and then lysed with nuclei permeabilized but still intact. DNA was then restricted with MboI restriction enzyme and the overhangs filled in incorporating a biotinylated base. Free ends were then ligated together in situ. Crosslinks were reversed, the DNA was sheared to 300-500bp and then biotinylated ligation junctions were recovered with streptavidin beads.

Standard Illumina library construction protocol. Briefly, DNA was end-repaired using a combination of T4 DNA polymerase, E. coli DNA Pol I large fragment (Klenow polymerase) and T4 polynucleotide kinase. The blunt, phosphorylated ends were treated with Klenow fragment (3' to 5' exo minus) and dATP to yield a protruding 3- 'A' base for ligation of Illumina's adapters which have a single 'T' base overhang at the 3’ end. After adapter ligation DNA was PCR amplified with Illumina primers for 14 cycles and library fragments of 400-600 bp (insert plus adaptor and PCR primer sequences) were purified using SPRI beads. The purified DNA was captured on an Illumina flow cell for cluster generation. Libraries were sequenced on the NextSeq500 following the manufacturer's protocols.

Point 2: It would be great if a flow chart should be provided for an easy understanding of how this entire study was performed.

Response 2: Given the materials and methods have been amended with further details as requested, we believe this isn’t necessary anymore.

Reviewer 2 Report

Dear authors,

Legumes represent very important part of  sustainable agriculture, because they can be used instead of artificial minerals, and moreover they improve the health of soil in the age of pesticide overusing.

Your article is very interesting and written in excellent way. All information is supported with the up-to-date references. Methodology is adequate, all the software you used is described and I see there was a lot of work with it, what I appreciate.  The obtained results are supported with easy-understandable figures and tables with rich description. I have some additional questions.

Question 1: Do you think that if your RNA-Seq libraries were derived from another part of plant (you wrote you used root and leaves), e.g. flowers, your annotations would be more complete like those you present in the article?

Question 2: Do you plan to do the similar research with other legumes or other trait of M. truncatula?

Author Response

Response to Reviewer 2 Comments

Legumes represent very important part of sustainable agriculture, because they can be used instead of artificial minerals, and moreover they improve the health of soil in the age of pesticide overusing.

Your article is very interesting and written in excellent way. All information is supported with the up-to-date references. Methodology is adequate, all the software you used is described and I see there was a lot of work with it, what I appreciate.  The obtained results are supported with easy-understandable figures and tables with rich description. I have some additional questions.

Point 1: Do you think that if your RNA-Seq libraries were derived from another part of plant (you wrote you used root and leaves), e.g. flowers, your annotations would be more complete like those you present in the article?

Response 1: RNA-seq data was used as an additional resource for annotation of the genome in combination with ab initio gene predictions from RNA-Seq alignments and evidence from protein homology studies that were used for annotation via the BRAKER2 pipeline and EvidenceModeler. Our opinion is that there might be expression difference in the genes and probably expression of certain floral specific transcripts if floral RNAseq libraries were chosen. We did not have access to the floral RNAseq libraries and the limitation is considered by using different methodologies in the annotation process as described before. We believe that R108 HiC assembly annotation is complete due to a variety of pipelines invoked to appropriately classify the annotations. We furnished these details in Materials and methods section as well as in the discussion section of the manuscript.

Point 2: Do you plan to do the similar research with other legumes or other trait of M. truncatula?

Response 2: We currently have another manuscript in preparation, which focuses on other legumes with similar research, which isn’t really the focus of this paper.
